# Food Grinding Behavior: A Review of Causality and Influential Factors

**DOI:** 10.3390/ani14131865

**Published:** 2024-06-24

**Authors:** Hao Tang, Wei-Wei Ge, Wan-Hong Wei, Sheng-Mei Yang, Xin Dai

**Affiliations:** 1College of Bioscience and Biotechnology, Yangzhou University, 48 East Wenhui Road, Yangzhou 225009, China; mx120221091@stu.yzu.edu.cn (H.T.); mx120221047@stu.yzu.edu.cn (W.-W.G.); whwei@yzu.edu.cn (W.-H.W.); smyang@yzu.edu.cn (S.-M.Y.); 2Jiangsu Co-Innovation Center for Prevention and Control of Important Animal Infectious Diseases and Zoonoses, Yangzhou University, Yangzhou 225009, China

**Keywords:** food grinding, causality, influential factors, functions, rodent

## Abstract

**Simple Summary:**

Food waste from grinding by experimental animals results in excess food scraps in cages, similar to how wild animals gnaw on vegetation and seeds, damaging the ecological environment. Despite food grinding’s function being partially understood, its biological purposes and influencing factors remain largely unexplored. This paper aims to review and explain potential causes of food grinding in animals, identify influencing factors, and discuss limitations. It emphasizes recent progress on gut microbiota’s significance in food grinding. The findings promote comprehensive food grinding research, benefiting laboratory animal husbandry and ecological protection.

**Abstract:**

Food waste is a common issue arising from grinding of food by experimental animals, leading to excessive food scraps falling into cages. In the wild, animals grind food by gnawing vegetation and seeds, potentially damaging the ecological environment. However, limited ecology studies have focused on food grinding behavior since the last century, with even fewer on rodent food grinding, particularly recently. Although food grinding’s function is partially understood, its biological purposes remain under-investigated and driving factors unclear. This review aims to explain potential causes of animal food grinding, identify influencing factors, and discuss contexts and limitations. Specifically, we emphasize recent progress on gut microbiota significance for food grinding. Moreover, we show abnormal food grinding is determined by degree of excess normal behavior, emphasizing food grinding is not meaningless. Findings from this review promote comprehensive research on the myriad factors, multifaceted roles, and intricate evolution underlying food grinding behavior, benefiting laboratory animal husbandry and ecological environment protection, and identifying potential physiological benefits yet undiscovered.

## 1. Introduction

Laboratory-reared mice eat large chunks of food, which occasionally fall into the cage; however, some mice grind their food, resulting in tiny pieces of food waste on the cage floor, which is known as food crumbs or residues (orts) [1]. Food grinding is also known as food spilling, food sorting, food fragmenting, or food wasting [2,3,4]. This behavior has provoked the interest of researchers because the presence of food crumbs could result in an overestimation of real food intake, which could pose a challenge in studies related to dietary restriction [5,6], dose calculations [7,8], metabolism, and assimilation efficiency, or other studies where the quantity of actual food consumption is crucial [2]. Moreover, this topic has a rich scientific history dating back to Barnes et al. in 1968, who observed an elevated food spillage in malnourished rats [9]. Food grinding is considered an abnormal and stereotypic or compulsive behavior in laboratory mice and rats and in other rodents kept in the laboratory [4], and it occurs even when high-quality and fairly homogenized food is provided [10]. Food grinding has also been observed in wild rodents, such as *Gerbillurus paeba*, *Mastomys natalensis*, North American porcupines (*Erethizon dorsatum*), meadow voles (*Microtus pennsylvanicus*), and prairie voles (*Microtus ochrogaster*) [11,12,13]. Well-developed incisors, the presence of harder enamel on the leading edges of the upper and lower incisors, and the ability of the lower jaw to move back and forth are the physical factors that allow for food grinding in rodents [14]. Food grinding is of great importance for feeding laboratory mice and other rodents because of its association with food waste and food production. Food crumbs from grinding behavior may average 60–70% of food consumption by laboratory rodents, even exceeding consumption in some individuals [15], causing huge economic loss of rodent chow and environmental uncleanliness in laboratory rodent raising. In wild rodents, food grinding is a food-wasting behavior that causes severe damage to vegetation and seeds. Many herbivorous rodents gnaw on herbaceous plants, leaving behind large amounts of uneaten plant residues [16], which can have a significant impact on vegetation. From an ecological perspective, the production of food residues plays an important role in ecosystems and affects the relationship between plants and animals because the impact of herbivores on plant populations does not depend on the amount of plant material that is ingested but rather on how much is destroyed and consumed [16].

Despite the considerable scholarly attention dedicated to investigating this phenomenon and the widespread dissemination of seemingly indisputable conclusions regarding food grinding, a consensus on its underlying causes and the numerous influencing factors remains elusive. Therefore, this paper summarizes the research to date on rodent food grinding, with a focus on the reasons why rodents grind food and the factors that influence it, to facilitate further in-depth investigations into the causes, influencing factors, and biological functions of rodent food grinding.

## 2. Methodology

We have comprehensively studied the food grinding behavior and published two scholarly articles related to the influences of food restriction and gut microbiota on food grinding behavior in Brandt’s vole [15,17]. However, we found that the food grinding behavior is not sufficiently investigated, and its reasons and influential factors are complex and uncertain. Therefore, we planned and conducted an in-depth review titled “Food Grinding Behavior: A Review of Causality and Influential Factors”. Firstly, we searched for food grinding behavior-related articles in various search engines, including Web of Science, PubMed, SpringerLink, and Elsevier Science. The license or privilege of using these search engines was purchased by Yangzhou University. The search terms used included food grinding, food gnawing, food spilling, food sorting, food fragmenting, food wasting, rodents, food spillage, food pellets, and food hardness. Secondly, we collected the relevant articles and compared and analyzed these articles. We also demonstrated a distinctive and thorough search of the articles, which is based on the references cited in the examined articles. Finally, we summarized the potential causality and influential factors of food grinding behavior and depicted the graphs.

## 3. Reasons for the Rodent Food Grinding Phenomenon

Currently, several main viewpoints have been proposed to explain why food grinding occurs in rodents (Figure 1): (1) food grinding in animals is a manifestation of the optimal foraging theory; (2) food grinding is caused by reduced environmental diversity; (3) food grinding is a manifestation of rodent nature; (4) food grinding is hereditarily determined; (5) food grinding is a type of table manners; (6) food grinding is a self-reinforcing behavior; (7) food grinding is related to the nest-building propensity of rodents; and (8) food grinding is a contrafreeloading behavior.

### 3.1. Optimal Foraging Theory

Optimal foraging theory has been proposed to explain the causes of food grinding in rodents [18], and it predicts that grinding may be beneficial to mice because foragers, which select a subset of potential food items to maximize net energy intake per unit of foraging time, grind food to extract more valuable food components to maximize energy intake [4]. The energy content of food consumed by MF1 mice, derived from a crossbreeding of LACA and CS1 mice, far exceeds that of orts [18]. Kerley and Erasmus [12] discovered that the choice and consumption of seeds by *Gerbillurus paeba*, *Mastomys natalensis*, and *Mus minutoides* were linked to the energy and carbohydrate levels of the seeds. Moreover, this behavior is positively correlated with the rate of energy intake, namely, the net rate at which foragers maximize their energy intake. Rodents reared in the laboratory do not eat the entire food pellet; rather, they grind the pellet, take a portion of it, and discard the rest. In addition, when fed seeds, they do not ingest the complete seed but discard a portion of the structural components [19]. Pritchett-Corning et al. [20] found that the energy content of orts produced by grinding is lower than that of pelleted food, suggesting that the mice choose to ingest the more energetic portion of the food. In Brandt’s voles (*Lasiopodomys brandtii*), the amount of food subjected to grinding does not significantly positively correlate with amount of food consumed, suggesting that Brandt’s voles consume the most energetic or optimal portion of the food so that they can obtain sufficient energy supplementation, even if the total amount consumed is reduced, which supports the hypothesis that food grinding is partially driven by optimal foraging strategies [15]. When the food supply is reduced, the amount of food grinding is decreased relative to the amount of food consumed by male Brandt’s voles [17], which further supports the optimal foraging strategy hypothesis for food grinding. When the food supply is reduced below the average food intake and the body mass growth rate is negative, food grinding further reduces the amount of available food to levels that cannot meet the daily energy demand. Additionally, food grinding is an energy-consuming behavior. Thus, no advantage would be gained by choosing food selectivity or food grinding. Small mammals can regulate their energy budget in response to a decrease in the food supply to manage periods of food shortage [21]. Therefore, voles reduce food grinding and ingest as much food as possible to maximize energy intake and reduce energy expenditure.

However, this hypothesis fails to explain the occurrence of this behavior in the presence of high-quality, uniformly processed food. Furthermore, it does not explain the selective manifestation of intensified food grinding behavior in certain individuals of the same rodent species.

### 3.2. Reduced Environmental Diversity

Natural conditions in the wild provide rodents with rich environments, such as expansive terrain and rich vegetation; however, laboratory rearing environments lack richness and are more homogeneous. Olsson and Dahlborn [22] concluded that a decrease in environmental diversity causes wild mice to discard normal activities, such as exploring and hiding, and induces stereotypic or compulsive food grinding behavior. The availability of wood blocks for gnawing decreases the amount of food spillage produced by sleep-deprived rats [23]. However, Cameron and Speakman [18] found that increasing the environmental diversity in mouse cages, such as by wooden blocks, metal blocks, and plastic toys, led to an increase in activity but did not reduce food grinding in mice. Similarly, laboratory rats housed in enriched cages (toys added) did not show significant reductions in food spillage compared to rats housed in non-enriched cages [24]. These contradictory findings suggest that environmental diversity may not be the main or key factor underlying food grinding behavior.

### 3.3. Rodent Nature

Rodents are naturally inclined to destroy the outer shell of seeds by gnawing to access the inner kernel, and they ultimately discard the seed shell [20]. This gnawing and discarding behavior strongly resembles food grinding, suggesting that the phenomenon of food grinding in rodents may be a direct result of the animal’s nature [20]. Additionally, scholars have proposed that rodent gnawing is a recognized stereotypical oral behavior and food grinding is a stereotypical gnawing behavior, indicating that food grinding is likely instinctive in rodents [25]. However, this hypothesis cannot account for the variations in food grinding behavior among individuals of the same rodent species, with some individuals exhibiting it frequently and others exhibiting it rarely.

### 3.4. Hereditary Determination

Although food grinding is highly variable among individual mice (the proportion of ground food particles relative to ingested food ranges from 2% to 40%), the phenomenon of food grinding is highly consistent within individuals. When offered a cellulose-based diet, ort production by two mice strains (MF1 and C57BL/6) differed significantly between individuals [18]. The extent of food grinding was highly correlated among siblings, and differences in food grinding were more pronounced across mouse strains. For example, the highest ort production averaged 4.30 g/day for MF1 mice and 8.41 g/day for C57BL/6 mice, which represented 13.3% and 43.3% of food intake, respectively [18]. Significant differences in food grinding were observed among replicate lines of mice after as few as 10 generations following their separation [3]. Differences in food spillage between mouse lines selected for high or low apparent food intake have also been observed [10]. Taken together, these results suggest that food grinding is heritable. Certain genes are likely responsible for governing the food grinding behavior, which may be influenced by either a single gene or a polygenic control mechanism.

However, the mechanisms underlying the inheritance of food grinding remain unknown. To validate this hypothesis, subsequent research ought to undertake mating experiments pairing rodents with pronounced food grinding traits with those manifesting less pronounced ones. This will ascertain the frequency and magnitude of food grinding behaviors in the progeny, thereby elucidating the patterns of inheritance for genes associated with food grinding behavior. Moreover, such work should clarify whether and how genotypes vary among rodents subjected to different degrees of food grinding. Gene sequencing and comparing the genomes of other mammals that do not exhibit food grinding may aid in identifying the key or related genes responsible for food grinding in rodents.

### 3.5. Type of Table Manners

The act of grinding food is not simply a random behavior but rather a significant aspect of the eating process. A study on house mice found that the amount of food fragmented during grinding is directly proportional to the total amount of food consumed. Based on these findings, Koteja, Carter, Swallow, and Garland [3] suggested that the intensity of food grinding could be considered a quantitative measure of “table manners.” In cold-exposure experiments, male mice that ground their food achieved lower levels of maximum food consumption and experienced a greater loss of body mass [3]. Furthermore, food grinding negatively correlates with the total litter mass at weaning. Hence, observations of the table manners in mice can offer insights into the qualities of both individual mice and their families. Messy eating, characterized by the grinding of more food during meals, may indicate lower physiological abilities in certain traits that are potentially linked to reproductive success (Darwinian fitness) [26,27]. Therefore, if this behavior is heritable, messy eaters may be perceived as weaker reproductive competitors. However, since food grinding indicates lower fitness, why has this trait been preserved through evolution and natural selection? In the future, partner selection in rodents should be investigated to test the hypothesis that food grinding corresponds to reproductively weak competitors.

### 3.6. Self-Reinforcing Behavior

Food fragmentation, as measured in mice provided with access to a wheel, exhibited a positive correlation with the extent of wheel running [3]. Moreover, the percentage of food fragments increased following deprivation of wheel access. These findings suggest that heightened food fragmentation may serve as an alternative to wheel-running activity. During wheel running, an animal undergoes rapid variations in movement speed and direction, partly because of external forces, such as the momentum and inertia of the wheel. Similar to humans enjoying amusement park rides, this experience may be reinforcing, especially when it involves motion in the vertical plane, which would initially be novel and thus stimulating [28]. Therefore, food fragmentation as a substitute for engaging in wheel-running activities could potentially become a behavior that reinforces itself [3]. More experiments are needed to test the hypothesis of food grinding as a self-reinforcing behavior.

### 3.7. Nest Building

Alternatively, laboratory rodents, such as mice, may pulverize food pellets to obtain nesting material, and dissimilarities in the degree of food fragmentation may mirror variations in the inclination to construct nests [3]. However, this hypothesized driving factor for food grinding has not been sufficiently substantiated, and experimental corroboration is limited.

### 3.8. Contrafreeloading Behavior

Animals will work for “earned” food even though identical “free” food is easily obtainable from a nearby resource. This is called contrafreeloading, observed in various vertebrates including wild and laboratory rats and mice [29,30,31]. In mazes, animals often choose long indirect routes to food over shorter direct ones [32]. They will also search for food [33] or solve puzzles for food [34] instead of taking identical free food. Thus, we propose food grinding could be a contrafreeloading behavior—an energy-cost behavior where animals grind for “earned” food. Food deprivation negatively affects contrafreeloading [29,35], consistently inhibiting food grinding [17]. In future, experiments evaluating contrafreeloading activity’s influence on food grinding in animals with grinding behavior could test this hypothesis. Although the contrafreeloading hypothesis for food grinding contradicts the optimal foraging theory hypothesis, both of these hypotheses imply that food grinding is related to foraging strategy.

What is the purpose of food grinding, and why do the genes associated with it maintain a certain frequency in animal populations? Analyzing the evolutionary and selective aspects of its existence may provide new insights into the underlying causes or drivers of food grinding, which could potentially serve physiological functions beneficial to animals that remain undiscovered. If there indeed exists a specific purpose or motivation behind food grinding, it should not be dismissed as a behavior devoid of function. Stereotypes, defined as repetitive, unchanging, and seemingly purposeless behavioral patterns [36], suggest that food grinding can be considered a stereotypic behavior to some extent.

## 4. Factors Affecting Rodent Food Grinding

Numerous factors influence food grinding in rodents, encompassing the characteristics and composition of the food, the quantity and manipulability of the food, the duration of sleep, the aging of the animal, the intestinal flora, the level of stress experienced by the animal, the ambient temperature in the environment, and gene expression (Figure 2).

### 4.1. Hardness of Food Particles

The phenomenon of food grinding in laboratory rodents, especially in laboratory mice, is related to the hardness of food particles, with harder food corresponding to lower food grinding [37,38]. A significant negative correlation is observed between ort production and diet hardness in the MF1 and C57BL/6 mouse strains [18]. A target hardness of at least 2.5 kg/mm^2^ is desirable to minimize ort production. Chows containing high percentages of fat tended to fall below this critical hardness and were ground more by both MF1 and C57BL/6 mice. These authors also revealed that the utilization of high-quality firm food notably mitigated the influence of food waste during the assessment of dietary intake. An optimal hardness of food particles for mice has been identified, and values above or below this hardness range do not lead to the maximum food intake. In addition, differences in the grinding capacity of food particles have been observed in different rodent species. However, further tests on food grinding behavior associated with feeds with the same formulation but different hardness levels must be performed to verify these findings. Moreover, detrimental behavior associated with harder foods, such as greater energy spent on food grinding and increased damage to teeth, may lead rodents to reduce their food grinding behavior on harder foods.

### 4.2. Food Availability and Food Intake Efficiency

When mice are supplied with only 80% of their average daily food intake, ort production drops to zero [18]. In male Brandt’s voles, when the quantity of food is reduced to below the daily food intake level, food grinding decreases sharply [17]. Thus, when accessible food is limited, food grinding is reduced because it consumes additional energy and wastes food, thereby reducing the animal’s energy intake. In contrast, an increase in the amount of food fed to mice corresponds to an increase in the amount of orts produced by food grinding [18]. This suggests that when more food is available, mice select the most nutritious portion of the food by grinding [18]. Kerley and Erasmus [12] found that the grinding of seeds by mice positively correlated with seed ingestion efficiency, energy production, and energy uptake rate and negatively correlated with seed handling time and seed size. This suggests that food grinding by animals increases in the presence of an adequate food supply or higher food intake efficiency. This also suggests that wild rodents cause more food wastage during seasons or years when food is abundant, which is a phenomenon that deserves further attention and validation. Based on this perspective, the food supply should be limited to slightly above the daily intake amount to minimize food waste generated by grinding food in laboratory rodents.

### 4.3. Ingredients in Food

Kerley and Erasmus [12] found that the consumption and grinding of seeds by mice were not related to either the polyphenol or free water content of the seeds and positively correlated with the soluble carbohydrate content of the seeds. The lipid, protein, and calorific yields were concentrated in the portion of the seed ingested by the mice, and the cell walls and ash, which contain woody accessory plant parts and seed coats, were avoided. Ort production and food grinding significantly increased as the proportion of cellulose and fiber in the diet increased [18]. Consistently, mice on a chow diet spilled more than 50% of their food onto the cage floor, whereas mice on a high-fat high-sucrose diet spilled approximately 25% [39]. Mice are known to dissect foods such as seeds to avoid unpalatable or toxic components; thus, they may discard up to 95% of the mass of processed food [12]. Thus, when presented with palatable food, animals will likely eat a greater proportion of the food and spill less than when presented with less palatable foods. The spillage of food in rats depends on the levels of thiamine in their diet, as a deficiency of thiamine led to a noticeable increase in food grinding behavior [40]. Further extensive research is required to identify the food constituents that impact food consumption. These constituents may contain essential nutrients, vitamins, and other undesirable compounds. Furthermore, the relationship between food nutrients and food grinding supports the notion that food grinding is driven by optimal foraging theory.

### 4.4. Length of Sleep

Sleep deprivation has been shown to increase food spillage in rats [41]. In sleep-deprived rats, a small amount of food spillage occurs at 24 h and is maintained for up to 96 h of sleep deprivation, even when a wooden block is provided [23]. Thus, sleep deprivation may increase stereotypical gnawing behavior rather than food demand [41,42], which could be related to an increase in dopaminergic D_2_ receptors in the brain [43]. This indicates that sleep duration influences the expression of genes in the brain, thereby affecting food grinding. This reinforces the hypothesis that food grinding behavior is genetically regulated.

### 4.5. Aging of Animals

Starr and Saito [2] found that elderly mice (aged 20 to 29 months) ground a greater quantity of food (36 ± 8%) than younger mice (aged 2 to 13 months) (18 ± 5%). Researchers have proposed that aging mice may experience a decline in sensory function, which affects their ability to accurately determine the nutritional value of food components. As a result, the stimulation of food grinding in these mice may be influenced by the principles of optimal foraging theory. Furthermore, a 20% dietary restriction did not significantly decrease the amount of food grinding in old C57BL/6 mice (20 months) [2]. This finding contradicts the expectations based on optimal foraging theory, which suggests that food restriction should effectively reduce the amount of ground food. Consequently, this contradiction implies that the motivation behind food grinding in aged rodents may not be solely to maximize energy intake.

### 4.6. Intestinal Flora of Animals

Our previous study found significant differences in the beta diversity of the fecal microflora of male Brandt’s voles (*Lasiopodomys brandtii*) exhibiting different food grinding intensities [15]. The genera *Alistipes* and *Aerococcus* may promote food grinding, whereas the genera *Carnobacteriaceae*, *Streptococcaceae*, *Atopostipes*, *Unclassified Clostridiaceae bacterium GM1,* and *Paenalcaligenes* may inhibit it [15]. Short-chain fatty acid (SCFA) acetate (acetate), a metabolite of the gut flora, may promote food grinding [15]. In a subsequent food restriction study in male Brandt’s voles, the abundance of *Unclassified Clostridiaceae bacterium GM1* and *Aerococcus* and content of acetate strongly related to food grinding [17]. SCFAs are important mediators in the gut microbe–brain axis pathway [44,45], and gut microorganisms can influence the host’s feeding behavior and appetite through SCFAs [46], suggesting that gut flora can influence the food grinding behavior of animals through their metabolites (Figure 3). However, the alpha diversity did not differ significantly between groups of Brandt’s voles with different intensities of food grinding, and only slight differences in the gut microbial community structures were observed, which implies that differences in the degree of food grinding in Brandt’s vole do not require large-scale alterations of gut microbes but rather may only require differences in part of the flora [15]. To elucidate the contribution of gut flora to food grinding behavior, further in-depth studies are needed, such as bacterial transplantation experiments or bacterial metabolite addition experiments. Such work may identify and verify the key flora and metabolites associated with food grinding.

### 4.7. Stress in Animals

Food grinding may be an abnormal behavior stimulated by stress in rodents [23]. Elevated food spillage, which is indicative of a broader behavioral anomaly, can be perceived as a display of heightened arousal in malnourished rats under stressful circumstances [9]. In open field experiments, the total distance traveled, time spent in the center area, and number of entries to the center area were decreased while the time spent stationary was increased in Brandt’s voles that exhibited strong food grinding. In addition, the abundance of the branching fungal genus *Alistipes*, which is associated with anxiety, was also higher in the guts of strong food grinding rats, indicating that Brandt’s voles in the strong food grinding group were more anxious than those in the weak food grinding group [15]. Sleep deprivation in rodents results in a heightened stress response, leading to increased gnawing behavior and food spillage. This behavior is believed to be a result of the complex interplay between the hypothalamus–pituitary–adrenal (HPA) axis and the dopaminergic system [23] (Figure 3). Alternatively, stereotypical behaviors are thought to serve as coping mechanisms to reduce stress [23]. In the future, the potential role of food grinding behavior in stress management should be investigated. If food grinding reduces stress, then coping with stress would be considered one of its functions. Therefore, it should be noted that stress is a factor that influences food grinding.

### 4.8. Gene Expression

Interestingly, a marked increase in food grinding behavior was observed in mice with hypothalamic neuropeptide Y1 receptor knockout, and the amount of ground food spilled on the cage floor was significantly elevated compared to that in the controls [47]. The ablation of Y1 receptors in the hypothalamus has been suggested to cause altered responses to masticatory trigeminal neurons in the brainstem, which are important for jaw movement, thereby resulting in increased food grinding (Figure 3). Dynorphin knockout mice also exhibit a marked food grinding phenotype when fed a chow diet [39]. Dynorphin knockout has been suggested to alter the perception of palatability and reduce the perceived palatability of chow food; alternatively, the lower basal serum glucagon levels in dynorphin knockout mice may contribute to the enhanced food grinding behavior of these animals. Given that the effects of dynorphins on feeding occur partially via interactions with the NPYergic system in the hypothalamic arcuate nucleus and that the expression of dynorphins and neuropeptide Y (NPY) are reciprocally reinforced, the dynorphin and NPY systems in the hypothalamus might play important roles in food grinding behavior (Figure 3). The Y1 receptor is implicated in the regulation of feeding behavior [48]. Consequently, the detected relationship between Y1 receptor and dynorphin with food grinding also indicates that food grinding is a feeding-related behavior. Unilateral medial forebrain bundle damage, which causes an obvious reduction in the levels of dopamine and norepinephrine in the ipsilateral mesolimbic area or striatum, increases food spillage in male Wistar albino rats, indicating that stereotypical food grinding behavior may be associated with dopamine pathways in the striatum [23,49]. These results indicate that food grinding behavior is genetically regulated in addition to its inheritance. Therefore, genes in the nervous pathway that regulate feeding and dopamine pathways in certain parts of the brain likely participate in the regulation of food grinding behavior (Figure 3). Such findings provide a basis for considering food grinding as an abnormal behavior because gene knockout or damage to certain brain areas consistently induces increased food grinding. Additionally, probing into the influence of feeding regulatory mechanisms and the dopaminergic pathways in the brain will enhance our understanding of the intricate molecular and genetic architecture that governs this behavior.

### 4.9. Other Factors

Additional factors such as the dimensions (length and diameter) of food particles can also affect food wastage and utilization in animals [38]. The smaller diameter (9.5 mm) pellets caused greater wastage in relation to hardness than with those of 12.7 mm diameter. Koteja, Carter, Swallow, and Garland [3] discovered that the occurrence of food grinding in mice was not linked to sex but rather to body weight or apparent digestibility. In contrast, our previous study did not identify an association between body weight and food grinding in Brandt’s voles, indicating that body weight is not a significant factor influencing or determining the extent of food grinding [15]. The ambient temperature can influence food grinding, as mice tend to fragment food less frequently in colder environments [3]. Further in-depth research will likely identify additional factors that affect food grinding.

## 5. Discussion

This review provides a comprehensive analysis of the driving, motivating, and influencing factors that underlie food grinding behavior. The “optimal foraging theory” has been considered by many researchers as the most plausible explanation for this behavior, a perspective that has been supported by several studies [4,15,18]. However, it is worth noting that more direct and substantial evidence is required to fully validate this theory. At present, there is no consensus among experts on the precise causalities of food grinding behavior. Most current theories are only in the preliminary stages of speculation and lack systematic and profound exploration.

If this behavior is motivated by the optimal forging theory, the nature of rodents, nest-building, the contrafreeloading hypothesis, or table manners, then food grinding itself cannot be considered a functionless behavior. Evidence from gene-knockout experiments and brain damage and stress studies indicates that food grinding could be an abnormal activity [39,47]. We suggest that food grinding should not be considered a normal or abnormal activity; rather, its intensity or degree should be described as either normal or abnormal in different animals. However, the threshold of abnormality is difficult to define because the intensity or degree of food grinding may be affected by multiple factors, such as the animal species or strain and food type, composition, and hardness. Intestinal flora is a hot topic in current research, and our previous study reported that intestinal flora may be related to food grinding behavior [15,17], thus providing new internal influencing factors or mechanisms for food grinding. Given the diversity of the functions of the intestinal flora, especially its effect on behavior, these microbes will represent a new direction in the study of food grinding and help clarify this behavior.

This behavior has energy costs; therefore, the balance between energy input and expenses is likely considered among animals that exhibit food grinding. According to optimal forging theory and contrafreeloading hypothesis, this behavior is mainly related to feeding. However, the biological function of food grinding has not been sufficiently investigated; thus, an evolutionary explanation of this behavior is lacking. The growth rate of body mass declined in tandem with the reduction in the relative amount of ground food and the ground-to-consumed food ratio in male Brandt’s voles, implying that food grinding potentially offers additional advantages or biological functions for voles beyond the mere maximization of energy intake [17]. Moreover, given the multifaceted nature of the reasons and influencing factors behind food grinding behavior, the underlying mechanisms are likely to be sophisticated and entwined, posing significant challenges in achieving a comprehensive understanding of this behavior.

## 6. Conclusions

Researchers have dedicated considerable effort to understanding the multifaceted reasons and factors underlying the intricate phenomenon of food grinding behavior. This area of study has been continually enriched and refined through an array of rigorous experiments since the last century. The reasons for food grinding are diverse and encompass an array of elements, including foraging, environmental stimuli, hereditary traits, innate animal instincts, table manners during the eating process, self-reinforcing habits, and the impulse to engage in nest-building activities. The factors that influence food grinding behavior in animals are equally complex and numerous, spanning from the availability and hardness of food to its nutritional composition, the effects of sleep duration, stress levels, aging processes, the intricate workings of intestinal flora, and even extending to the realm of genetic expression. However, pinpointing the precise causes, understanding the multifaceted influencing factors, and elucidating the functions of animal food grinding presents a formidable challenge. These tasks demand further in-depth exploration, particularly with regard to their nuanced functions. This specific area of research holds tremendous value in fostering a deeper comprehension of the myriad factors, multifaceted roles, and intricate evolution underlying food grinding behavior. Additionally, these insights will undoubtedly aid in optimizing laboratory feeding regimens for rodents, enhancing assessments, and improving predictions of wild rodent damage to vegetation and ecological systems.

## Figures and Tables

**Figure 1 animals-14-01865-f001:**
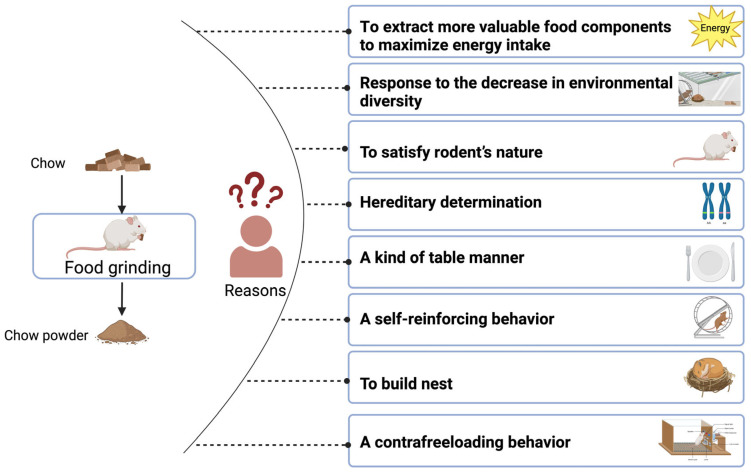
The potential causality of food grinding behavior.

**Figure 2 animals-14-01865-f002:**
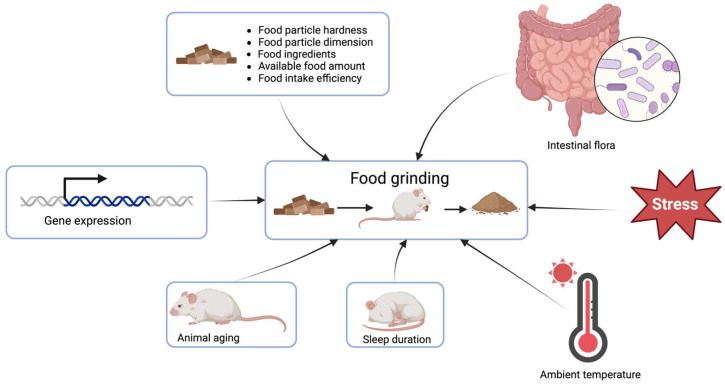
The potential influential factors of food grinding behavior.

**Figure 3 animals-14-01865-f003:**
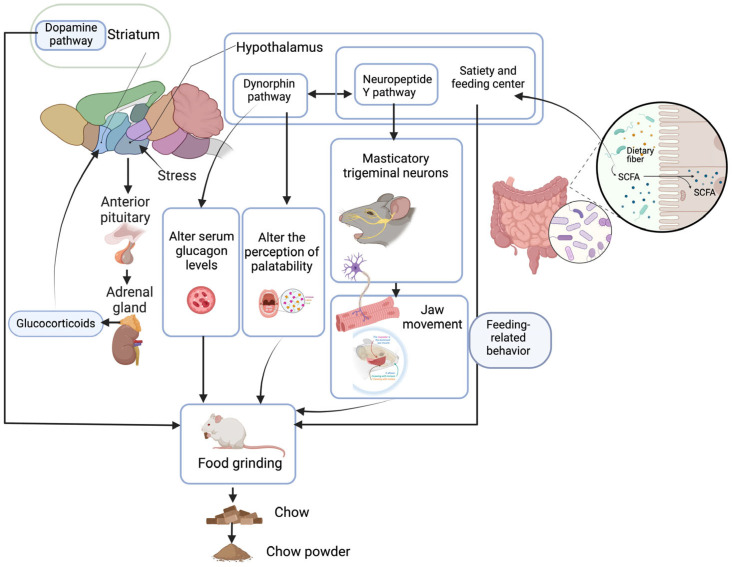
The potential neural and molecular mechanism underlying food grinding behavior.

## Data Availability

Data sharing not applicable.

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
