# Peer review of "Food Grinding Behavior: A Review of Causality and Influential Factors"

_animals, 2024, doi:10.3390/ani14131865_

Round 1

Reviewer 1 Report

Comments and Suggestions for Authors

I read this review of studies on food-grinding behavior with great interest. Being able to review a manuscript without finding typographical or conceptual errors was a rare experience —my congratulations to the authors—.

The content of the review is accurate. I have not found any contradictions or serious omissions. However, I suggest that the authors consult the scientific literature on "contrafreeloading" (N>60 papers in the Scopus database, N>80 papers in the Web of Science database). Contrafreeloading is a pattern of behavior described as paradoxical and not yet fully explained (Osborne 1977, Inglis et al. 1997). This pattern consists of using costly forms of feeding even when less costly alternatives are available. Among the functions of contrafreeloading, some have been proposed that could be applied to food grinding behavior. Therefore, contrafreeloading may be already in some of the viewpoints in the manuscript, although it is not explicitly cited. Furthermore, the concept of contrafreeloading may be known to the authors. Note that contrafreeloading i) is not a self-reinforcing behavior; ii) it challenges the Optimal Foraging Theory (OFT) hypothesis; iii) it may be a response to an oversimplified environment. Including contrafreeloading in the review is not mandatory; it is only a recommendation and should be considered as such by the authors.

Inglis, I.R., Forkman, B. & Lazarus, J. (1997) Free food or earned food? A review and fuzzy model of contrafreeloading. Animal Behaviour, 53: 1171-1191.

Osborne, S.R. (1977) Free food (contrafreeloading) phenomenon - review and analysis. Animal Learning & Behavior, 5: 221-235.

Author Response

Comment: The content of the review is accurate. I have not found any contradictions or serious omissions. However, I suggest that the authors consult the scientific literature on "contrafreeloading" (N>60 papers in the Scopus database, N>80 papers in the Web of Science database). Contrafreeloading is a pattern of behavior described as paradoxical and not yet fully explained (Osborne 1977, Inglis et al. 1997). This pattern consists of using costly forms of feeding even when less costly alternatives are available. Among the functions of contrafreeloading, some have been proposed that could be applied to food grinding behavior. Therefore, contrafreeloading may be already in some of the viewpoints in the manuscript, although it is not explicitly cited. Furthermore, the concept of contrafreeloading may be known to the authors. Note that contrafreeloading i) is not a self-reinforcing behavior; ii) it challenges the Optimal Foraging Theory (OFT) hypothesis; iii) it may be a response to an oversimplified environment. Including contrafreeloading in the review is not mandatory; it is only a recommendation and should be considered as such by the authors. Inglis, I.R., Forkman, B. & Lazarus, J. (1997) Free food or earned food? A review and fuzzy model of contrafreeloading. Animal Behaviour, 53: 1171-1191.Osborne, S.R. (1977) Free food (contrafreeloading) phenomenon - review and analysis. Animal Learning & Behavior, 5: 221-235.

Response: Thanks for your careful reviewing and valuable suggestion. According to your advice, we have carefully examined the elements of contrafreeloading in food grinding behavior and considered that contrafreeloading could be applied to food grinding behavior. We have added the contrafreeloading as a reason for food grinding behavior in line 114 to 115 as “and (8) food grinding is a contrafreeloading behavior”, “a contrafreeloading behavior” in figure 1, in line 238 to 250 as “2.8. Contrafreeloading Behavior Animals will work for “earned” food even though identical “free” food is easily ob- tainable from a nearby resource. This is called contrafreeloading, observed in various vertebrates including wild and laboratory rats and mice[29-31]. In mazes, animals often choose long indirect routes to food over shorter direct ones[32]. They'll also search for food [33] or solve puzzles for food [34] instead of taking identical free food. Thus, we propose food grinding could be a contrafreeloading behavior - an energy-cost behavior where animals grind for “earned” food. Food deprivation negatively affects contrafreeloading[29,35], consistently inhibiting food grinding[20]. In future, experiments evaluating contrafreeloading activity's influence on food grinding in animals with grinding behavior could test this hypothesis. Although the contrafreeloading hypothesis for food grinding contradicts the optimal foraging theory hypothesis, both of these hypotheses imply that food grinding is related to foraging strategy.”, and in line 433 and 448 as “contrafreeloading hypothesis”. The cited references also have been added to the part of references in our manuscript.

Reviewer 2 Report

Comments and Suggestions for Authors

This is a well-structured and written review on food grinding in rodents, possible causes and solutions. I believe it is a meaningful contribution for the discussion about the motivation of animals and factors contributing for this phenomenon.

I would suggest adding economical impact as a consequence of food waste from grinding, which can impact animal facility. 

I also suggest that all text in fig. 3 is horizontal. Makes no sense to have it in different orientations. 

Author Response

Comment 1: I would suggest adding economical impact as a consequence of food waste from grinding, which can impact animal facility. 

Response: Thanks for your careful reviewing and valuable advice. According to your suggestion, we have added the economic impact of food grinding on laboratory rodents raising in line 74 to 77 as “Food crumbs from grinding behavior may average 60%-70% of food consumption by laboratory rodents, even exceeding consumption in some individuals [15], causing huge economic loss of rodent chow and environmental uncleanliness in laboratory rodent raising.”

Comment 2: I also suggest that all text in fig. 3 is horizontal. Makes no sense to have it in different orientations. 

Response: According to your suggestion, we have revised the figure 3 and have changed the orientation of all text to horizontal.